# Anticoagulation-related complications and their outcomes in hemodialysis patients with acute kidney injury at selected hospitals in Ethiopia

Hanan Muzeyin Kedir[1]* , Abdella Birhan Yabeyu[2], Addisu Melkie Ejigu[3], Tamrat Assefa Tadesse[1], Eskinder Ayalew Sisay[1]

1 Department of Pharmacology and Clinical Pharmacy, School of Pharmacy, Collage of Health Sciences, Addis Ababa University, Addis Ababa, Ethiopia, 2 Department of Pharmacy, College of Medicine and Health Sciences, Ambo University, Ambo, Ethiopia, 3 Department of Internal Medicine, Addis Ababa University, Addis Ababa, Ethiopia

☯ These authors contributed equally to this work.
* hanan.muzeyin@aau.edu.et

**Data Availability Statement:** All relevant data are within the manuscript and its Supporting Information files.

## Abstract

### Introduction

During hemodialysis (HD), the presence of clots in the dialyzer can diminish the effective surface area of the device. In severe cases, clot formation in the circuit can halt treatment and lead to blood loss in the system. Thus, ensuring proper anticoagulation during HD is crucial to prevent clotting in the circuit while safeguarding the patient from bleeding risks. This study aimed to evaluate anticoagulation outcomes and related factors in HD patients with acute kidney injury (AKI) at selected hospitals in Ethiopia.

### Method

A prospective, multicenter observational study was carried out between October 1, 2021, and March 31, 2022. The study encompassed all AKI patients undergoing HD at least once during the study period. Descriptive statistics were utilized to summarize the data, and multinomial logistic regression analysis was employed to identify factors associated to clotting and bleeding.

### Results

Data were gathered from 1010 HD procedures conducted on 175 patients. Extracorporeal circuit clotting was detected in 34 patients during 39 (3.9%) dialysis sessions while bleeding incidents occurred in 27 patients across 29 (2.9%) sessions. A statistically significant association was found between both the total number of HD treatments and blood flow rate with incidents of clotting. Factors such as length of hospitalization, serum creatinine levels at admission, signs and symptoms associated with uremia, along utilization of anticoagulants or antiplatelet medications demonstrated an association with bleeding events.

**Funding:** The author(s) received no specific funding for this work.

**Competing interests:** The authors have declared that no competing interests exist.

**Abbreviations:** AKI, Acute Kidney Injury; HD, Hemodialysis; SPHMMC, St. Paul's Hospital Millennium Medical College; TASH, Tikur Anbessa Specialized Hospital; UFH, Unfractionated heparin.

## Conclusion

Clotting affected 19.4% of participants, while bleeding occurred in 15.4%, underscoring the importance of close monitoring. The frequency of HD sessions and blood flow rate are correlated with clotting, while hospitalization duration, serum creatinine levels, uremic symptoms, and anticoagulant use are associated with bleeding events.

## Introduction

Hemodialysis (HD) constitutes a medical procedure that facilitates the removal of waste products from the bloodstream externally via a device outfitted with specialized filtration systems. The predominant cause for arteriovenous access failure is attributed to thrombosis, representing 80%-85% of cases [1,2]. Within the HD protocol, components such as the dialyzer and other elements of the extracorporeal blood circuit are also vulnerable to thrombotic events. This includes tubing, arterial and venous bubble traps in addition to needles or catheters utilized for vascular access—all factors contributing to coagulation phenomena [3,4]. Notably, arterial and venous bubble traps exhibit a heightened susceptibility towards clotting owing to diminished blood flow rates leading potentially to stasis and hence clot formation. Such premature cessation of HD sessions precipitates inadequate dialysis alongside possible hematologic losses. Furthermore clots within this circuitry can herald significant negative repercussions including a reduction in solute clearance during therapy amplified costs, and increased operational burden [5–7].

Maintaining an optimal balance between insufficient and excessive heparinization is essential for achieving proper anticoagulation during HD to avert bleeding and clot formation within the extracorporeal circuit, respectively [6,8]. It is imperative to administer the minimal efficacious dose of anticoagulant to ensure that the dialyzer and venous chamber remain clear of blood cell fragments while also facilitating rapid hemostasis at the vascular access site following treatment [5]. Although anticoagulant dosing varies among individuals due to patient-specific factors and HD-related variables, it is generally lower than doses required for full anticoagulation. Inadequate levels of anticoagulation may reduce dialysis's effectiveness in eliminating waste products from the bloodstream [4–6].

Unfractionated heparin (UFH) is extensively utilized as an anticoagulant during HD owing to its long-standing history in clinical practice and a desirable safety profile characterized by a brief half-life [8,9]. The European guidelines for optimal HD practices suggest an initial bolus of 50 IU/kg UFH administered via the arterial access needle, followed by a continuous infusion maintenance dosage ranging from 500 to 1500 IU of UFH per hour. It's recommended that the use of UFH be minimized, especially since patients with acute kidney injury AKI often have compromised health and may require daily dialysis treatments. In instances of clotting or bleeding, modifications in dosage or the implementation of heparin-free dialysis methods might become necessary. Although alternative anticoagulation techniques targeting the extracorporeal circuit exist, their adoption has been limited due to their complex nature and higher requisites for administration time and workforce involvement for monitoring purposes [10–12].

Critically ill patients often display impaired coagulation and low platelet counts, which significantly increases their risk of bleeding when systemic anticoagulants are used. As such, it is advised that these individuals avoid any treatments that further heighten this risk, especially those involving systemic anticoagulants [6,13]. In situations where the risk of bleeding is a

concern, the European Best Practice Guidelines recommend alternatives like regular saline flushing or local citrate anticoagulation during HD, avoiding systemic anticoagulants entirely. This approach not only facilitates quick adjustments to treatment but also enables easy monitoring for clotting within the dialyzer and may help in preventing clots from forming initially [2,10]. The goal of this study is to explore outcomes related to anticoagulation therapy and factors influencing its outcome among AKI patients receiving HD at Tikur Anbessa Specialized Hospital (TASH) and St. Paul's Hospital Millennium Medical College (SPHMMC) in Addis Ababa, Ethiopia.

## Material and methods

### Study setting

The study took place in the dialysis departments of TASH and SPHMMC, located in Addis Ababa, Ethiopia. TASH introduced HD services back in 1980 and presently caters to acute dialysis needs. Meanwhile, SPHMMC began its provision of acute dialysis services in August 2013 before extending these services to include maintenance dialysis for kidney transplantation starting in early 2015 [14].

Despite the lack of a specific written protocol for HD patients, anticoagulation was frequently administered at both hospitals. UFH was used during HD, but the prescribed doses of heparin varied slightly between the two centers. At TASH, UFH was administered in one of three ways: (i) the standard dose, 1 ml (5000 IU) of UFH, was diluted with 4 ml of normal saline (total of 5 ml). This was followed by a 2 ml initial bolus and a continuous infusion of 1 ml/hr; (ii) the minimal or half dose of 0.5 ml (2500 IU) of UFH was diluted with 4.5 ml of normal saline (total of 5 ml). This was followed by a 2 ml initial bolus and a continuous infusion of 1 ml/hr; (iii) heparin-free saline flushes of 30 to 50 ml were given every 30 minutes. At SPHMMC, 1 ml of UFH was diluted with 3 ml of normal saline (total of 4 ml) and administered as a bolus of 1 ml, followed by a continuous infusion of 1 ml/hr.

### Study design and period

A comprehensive, multicenter observational study was carried out over a duration of six months, starting on October 1st, 2021, and concluding on March 31st, 2022. This study took place at two major hospitals (TASH and SPHMMC) in Addis Ababa, Ethiopia.

### Sampling and sample size determination

On average, the dialysis clinics at TASH and SPHMMC serve approximately 15 and 30 AKI patients per month, respectively. All patients meeting the eligibility criteria were considered for inclusion in the study. Consequently, a total of 175 patients identified during the study period, meeting the established criteria, participated in this research. Specifically, 122 of these patients received treatment at SPHMMC, while the remaining 53 were from TASH.

### Inclusion and exclusion criteria

All patients suffering from AKI, aged 12 years and older, who underwent HD at least once during the designated study period were eligible for inclusion in this study. Nonetheless, individuals diagnosed with chronic kidney disease who were undergoing regular maintenance dialysis treatments, as well as those who expressed a refusal to participate in the study, were systematically excluded from consideration.

### Data collection instruments and techniques

The data collection questionnaire, consisting of four sections, was developed through a thorough review of relevant literature related to the study's objectives. The aim was to collect demographic characteristics and clinical variables of patients to assess outcomes related to anticoagulation. Pre-testing was carried out, and all necessary adjustments to the data collection tool were implemented [4,6,9,15–18].

The initial section of the data collection tool was designed to gather information regarding the sociodemographic characteristics of the patients. Subsequently, the second section assessed the clinical features of the patients, including inquiries about the cause of AKI, indications for HD, concomitant medical conditions, co-administered medications, and current use of anticoagulants and/or antiplatelet drugs. The third part of the data collection tool focused on recording essential laboratory results upon admission and discharge. Finally, the fourth section specifically documents the comprehensive details of each HD session. Patients were closely monitored during each session, with data collectors prospectively recording various parameters related to HD, such as venous pressure, ultrafiltration volume, blood flow, dialysate flow, and approach to UFH usage during HD. Additionally, incidents of clotting or bleeding were documented, along with the interventions implemented to manage these occurrences.

### Data quality assurance

To ensure data quality, the principal investigator closely monitored each activity to ensure comprehensive data collection. Unclear terms were promptly clarified, and the necessary corrections were made during the pre-test phase before commencing the main study. Data collection was performed using three BSc. nurses and a pharmacist who were recruited as data collectors. They underwent training to establish a consistent understanding and interpretation of the instrument, including a briefing on the study's objectives and commitment to maintaining core ethical principles throughout the data collection period.

### Data analysis

Data completeness was verified before entering the statistical package for social science (SPSS) version 26 for a thorough analysis. Descriptive statistics, including the mean, median, percentage, and standard deviation (SD), were then used to succinctly summarize the data. Multinomial logistic regression was used to identify predictive factors for anticoagulation-related outcomes in patients with AKI undergoing HD. A P-value $< 0.05$ was considered the threshold for declaring statistically significant associations.

### Ethical considerations

Ethical clearance was secured from Addis Ababa University, School of Pharmacy, the Ethical Review Committee (reference number ERB/SOP/257/13/2021), and the Institutional Review Board of SPHMMC (reference number PM25/386). Study participants provided both written and verbal consent before participating, and their confidentiality was maintained by avoiding the recording of identifying information such as patient names. The collected data remained securely stored throughout the study and was appropriately destroyed upon study completion.

### Operational definition

Anticoagulation-related complication: were defined as the occurrence of any circuit clotting or bleeding during HD, whether with the use of UFH as an anticoagulant for dialysis or in a heparin-free approach.

# Results

## Socio-demographic characteristics of study participants

Of the 175 patients included in the study, an almost equal number were males (50.9%) and females (49.1%), with a mean age of 40 ± 17.8 years involved and four females were pregnant. Among the 175 patients, 66.9% were paid out-of-pocket for dialysis and 2.3% were involved in smoking, drinking alcohol, and chewing khat (Table 1).

## Clinical characteristics

The average length of hospital stay was 14.2 days, range, of 2–40 days. The three most common causes of AKI were acute tubular necrosis (29%), nephrotic syndrome (24%), and sepsis (22.9%). Pregnancy-related causes also included preeclampsia/eclampsia (5.1%) and HELLP syndrome (4.6%). The major indications for dialysis were uremic signs and symptoms (70.3%), followed by fluid overload (50.3%). Common underlying comorbidities identified were hypertension (55.4%) and anemia (49.7%), and 13.1% of AKI cases were superimposed on chronic kidney disease. UFH 7500 IU twice per day (48%) was the most widely used anticoagulant regimen in the study population, followed by 5000 IU twice a day (14.3%) to prevent deep venous thrombosis (Table 2).

## Laboratory findings

Upon admission and upon discharge, it was noted that there were low hemoglobin levels present. Furthermore, the mean serum creatinine level recorded at the time of admission stood at 8.3 ± 3.4 mg/dl, which experienced a significant decrease by 3 mg/dl by the time of discharge. Table 3 in our document provides a summarized outline showing these changes among other selected laboratory parameter values observed during this period.

**Table 1. Socio-demographic characteristics of AKI patients who undergo HD (n = 175).**

| Variable | Category | n (%) |
|---|---|---|
| Age (years),mean ± SD | | 40±17.8 |
| Sex | Male | 89 (50.9) |
| | Female | 86 (49.1) |
| Marital status | Single | 55 (31.4) |
| | Married | 103 (58.9) |
| | Widowed | 10 (5.7) |
| | Divorced | 7 (4) |
| Pregnancy | Pregnant | 4 (4.7) |
| | Not pregnant | 71 (82.5) |
| | Postpartum | 11 (12.8) |
| Education | Unable to read and write | 23 (13.1) |
| | Primary education | 42 (24) |
| | Secondary education | 63 (36) |
| | Diploma and above | 47 (26.9) |
| Weight (Kg) | | 61±13.9 |
| Method of payment | Insurance | 58 (33.1) |
| | Out of pocket | 117 (66.9) |
| Substance use | Smoking | 7 (4) |
| | Chewing Khat | 9 (5.1) |
| | Alcohol | 26 (14.9) |
| | Smoking + Khat + Alcohol | 4 (2.3) |

**Table 2. Clinical characteristics of AKI patients who undergo HD (n = 175).**

| Variables | Category | n (%) |
|---|---|---|
| | | Yes |
| Length of hospitalization (days), mean ± SD | | 14.2 ± 9.9 |
| Cause of AKI | Acute tubular necrosis | 49 (28) |
| | Nephritic syndrome | 42 (24) |
| | Sepsis | 40 (22.9) |
| | Severe dehydration | 17 (9.7) |
| | Interstitial nephritis | 25 (14.3) |
| | Shock | 19 (10.9) |
| | Poison | 17 (9.7) |
| | Urinary tract obstruction | 11 (6.3) |
| | Nephrotoxic drugs | 8 (4.6) |
| | Rhabdomyolysis | 3 (1.7) |
| | HELLP syndrome | 8 (4.6) |
| | Post-kidney transplant rejection | 6 (3.4) |
| | Lupus nephritis | 9 (5.1) |
| | Preeclampsia/Eclampsia | 9 (5.1) |
| | Acute Glomerulonephritis | 7 (4) |
| | Hemorrhage | 5 (2.9) |
| | Others* | 4 (2.3) |
| Dialysis indication | Uremic signs and symptoms | 123 (70.3) |
| | Fluid overload | 88 (50.3) |
| | Hyperkalemia | 79 (45.1) |
| | Metabolic Acidosis | 66 (37.7) |
| Comorbidity | Atrial fibrillation | 1 (0.6) |
| | Coronary arterial disease | 1 (0.6) |
| | Stroke | 6 (3.4) |
| | Hypertension | 97 (55.4) |
| | Congestive heart disease | 22 (12.6) |
| | Gastrointestinal ulceration | 29 (16.6) |
| | Chronic Kidney Disease | 23 (13.1) |
| | Anemia | 87 (49.7) |
| | Deep vein thrombosis | 5 (2.9) |
| | Pulmonary embolism | 3 (1.7) |
| | Type 2 Diabetes | 22 (12.6) |
| | HIV | 11 (6.3) |
| | Pulmonary tuberculosis | 9 (5.1) |
| | Hepatitis B virus | 5 (2.9) |
| | Others** | 41 (23.4) |
| Concurrent anticoagulant and/or antiplatelet | | 124 (70.9) |
| | UFH 2500 IU SC BID | 1 (0.6) |
| | UFH 5000IU SC Bid | 25 (14.3) |
| | UFH 7500IU SC Bid | 84 (48) |
| | UFH 17500IU SC Bid | 5 (2.9) |
| | Warfarin 5mg PO Daily | 8 (4.6) |
| | ASA 81mg PO Daily | 20 (11.4) |
| | Enoxaparin 40mg SC Bid | 3 (1.7) |
| | Clopidogrel 75mg PO Daily | 2 (1.1) |

*(Continued)*

**Table 2.** (Continued)

| Variables | Category | n (%) |
| --- | --- | --- |
| | | Yes |
| Concomitant drugs use | | 166 (94.9) |

* (malaria, kidney trauma, cardio-renal syndrome, acute fatty liver of pregnancy).

** (prostate cancer, BPH, epilepsy, leukemia, gout arthritis, cerebral malaria).

## Characteristics of HD sessions

From October 2021 to March 2022, a total of 1010 HD sessions were conducted on 175 patients. The average frequency of dialysis sessions was calculated at 5.8 ± 3.8 per patient, ranging from one single session to a maximum of fifteen sessions with each session lasting an average duration of approximately 3.1 ± 0.6 hours. Patients underwent HD therapy thrice weekly using catheters as the primary means for vascular access across all cases in this study. All patients utilized a non-tunneled (temporary) HD catheter, constructed from single-use polyurethane, to establish a temporary blood connection for acute HD sessions. Catheters were inserted in either the jugular or femoral vein sites. During these treatments, blood products were administered in approximately 9.8%, or essentially, during almost ten out of all hundred HD procedures. UFH was administered for circuit anticoagulation in 626 sessions, accounting for 62% of the total. The average dose given was 1 mL (5000 IU). Of these UFH sessions, infusion was stopped 60 minutes before discontinuation in 546 cases, making up 87.2% of the total. The average pressure in the venous chamber was recorded as 125 ± 39.3 mmHg, with the dialysis blood flow rate averaging 227.2 ± 22.8 ml/min (Table 4).

## Anticoagulation-related complications and HD outcome

Of the 175 patients, extracorporeal circuit clotting occurred in 34 patients in 39 (3.9%) sessions and 27 patients in 29 (2.9%) experienced bleeding. HD circuit clotting resulted in the discarding of the bloodline and early termination of dialysis in 17 sessions. Additionally, nursing interventions, that is, the use of UFH and a normal saline flush, were required in 10 and 5 sessions, respectively, to prevent abrupt HD treatment interruptions due to circuit clotting. Some of the measures taken when bleeding occurred included holding UFH at the time, and for the next session/s (n = 14 sessions), blood transfusion (n = 2 sessions), and applying pressure at the catheter insertion site in 6 sessions (Table 5). On the other hand, the majority of AKI patients (44.6%) were discharged with improvement, while more than one-fourth (26.9%) of them died after undergoing HD (Fig 1).

**Table 3. Selected laboratory values of AKI patients who undergo HD (n = 175).**

| Variables | Mean ± SD | |
| --- | --- | --- |
| | Admission | Discharge |
| Hemoglobin (g/dl) | 10.3 ± 7.6 | 10.3 ± 6.4 |
| WBC count ($10^3$/ml) | 11.1 ± 5.6 | 9.8 ± 4.9 |
| Platelet count ($10^3$/ml) | 235.9 ± 140.1 | 268.3 ± 131 |
| Serum creatinine (mg/dl) | 8.3 ± 3.4 | 5.3 ± 3.1 |
| Urea (mg/dl) | 137.5 ± 80.3 | 82.1 ± 63.4 |
| Serum potassium (mEq/L) | 5.3 ± 1.1 | 4.4 ± 0.8 |

**Table 4. Characteristics of HD sessions included in the study (N = 1010).**

| Variables | | mean ± SD |
|---|---|---|
| Total number of HD sessions (N = 1010) | | 5.8 ± 3.8 |
| HD duration (in hour) | | 3.1 ± 0.6 |
| Venous chamber pressure (mmHg) | | 125 ± 39.3 |
| Ultrafiltration volume (ml) | | 885 ± 650.9 |
| Blood flow rate (ml/min) | | 227.2 ± 22.8 |
| Dialysate flow rate (ml/min) | | 481 ± 46.6 |
| Total UFH dose (ml)* | | 1 ± 0.2 |
| | **Category** | **n (%)** |
| UFH infusion stopped | | |
| | Before 30 minutes | 70 (11.3) |
| | Before 60 minutes | 546 (87.2) |
| | Before 120 minutes | 8 (1.3) |
| | At the end of dialysis | 1 (0.2) |

## Factors associated with anticoagulation-related outcomes

To determine the predictors of anticoagulation-related outcomes during HD (clotting and bleeding risks), multinomial regression analysis was used. Based on the model comparing participants who had HD circuit clotting with those who did not have any clotting/bleeding, as the total number of HD sessions increased, the probability of circuit clotting increased

**Table 5. Anticoagulation-related complication and HD outcome among AKI patients in the study.**

| Variables | | mean ± SD |
|---|---|---|
| Circuit clotting (34 patients, 19.4%) | | 39 (3.9) |
| Measures taken when circuit clotting occurs | Discard the bloodline and HD terminated early | 17 (1.7) |
| Blood transfusion during HD | | 99 (9.8) |
| Using UFH during HD | | 626 (62) |
| | Additional UFH given | 10 (1) |
| | Continued HD by using normal saline flush | 5 (0.5) |
| | Return the blood and HD terminated early | 2 (0.2) |
| | Manipulate the access and continued HD | 3 (0.3) |
| | Use UFH intermittently for the next sessions | 2 (0.2) |
| Bleeding (27 patients, 15.4%) | | 29 (2.9) |
| | Bleeding from nose | 15 (1.5) |
| | Catheter site bleeding | 6 (0.6) |
| | Vaginal bleeding | 4 (0.4) |
| | Others** | 4 (0.4) |
| Measures are taken when bleeding occurs | Hold UFH at the time and for the next session/s | 14 (1.4) |
| | Blood transfused | 2 (0.2) |
| | Applied pressure at the catheter insertion site | 6 (0.6) |
| | Use UFH intermittently for the next sessions | 2 (0.2) |
| | Return the blood and HD terminated early | 1 (0.1) |
| | No measure taken | 4 (0.4) |

*1ml = 5000 IU.

** bleeding from rectum, bloody vomiting, subcutaneous hematoma, bleeding from bullet penetrating injury site.

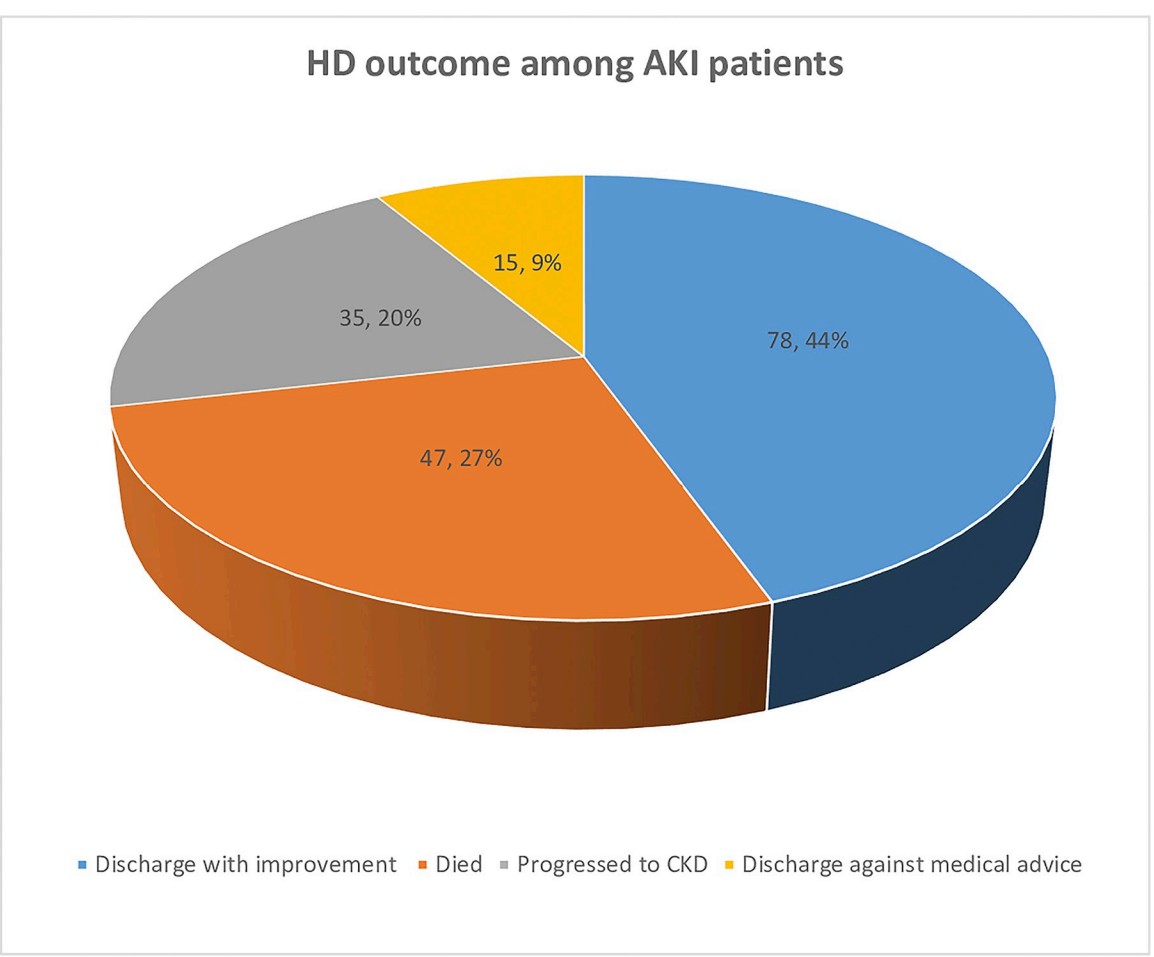

**Fig 1. HD outcome among AKI patients in the study.**

(AOR = 1.932, 95% CI, 1.227–3.043, p = 0.004). In contrast, a higher blood flow rate was associated with a lower HD circuit clotting risk (AOR = 0.868, 95% CI, 0.812–0.928, p = 0.001).

On the other hand, bleeding was more likely to occur in participants who had longer hospitalization and elevated serum creatinine at admission (AOR = 1.247, 95% CI, 1.053–1.478, p = 0.010; AOR = 1.886, 95% CI, 1.285–2.769, p = 0.001) respectively. While the rate of bleeding in patients who had no uremic signs and symptoms was lower (AOR = 0.092, 95% CI, 0.009–0.955, p = 0.004) than those having it. Likewise, patients who did not receive prophylactic or therapeutic anticoagulant and/or antiplatelet drugs were less likely to develop bleeding (AOR = 0.017, 95% CI, 0.001–0.446, p = 0.014) compared to those taking these drugs (Table 6).

## Discussion

As the first prospective study in the Ethiopian context on anticoagulation among patients with AKI on HD, this study provides insight into extracorporeal circuit clotting or bleeding occurrences and the predictive factors associated with these complications. In this study, 1010 HD sessions performed on 175 patients were prospectively followed. Clotting and bleeding data are expressed as percentages of both participants and sessions. Circuit clotting occurred in 3.9% of HD sessions (19.4% of patients), and bleeding in 2.9% of sessions (15.4% of patients).

**Table 6.  Multinomial regression of predictive factors associated with anticoagulation-related outcomes during HD.**

| Variables | Outcomes | | |
|---|---|---|---|
| | Clotting<br>AOR (95% CI) | Bleeding<br>AOR (95% CI) | P-value |
| Total no of HD sessions | 1.932 (1.227–3.043)[a] | 1.068 (0.713–1.601) | 0.004* |
| Blood flow rate (ml/min) | 0.868 (0.812–0.928)[a] | 1.026 (0.974–1.081) | <0.001* |
| Length of hospitalization (days) | 0.946 (0.843–1.06) | 1.247 (1.053–1.478)[a] | 0.010* |
| Serum creatinine (mg/dl) at admission | 0.911 (0.699–1.189) | 1.886 (1.285–2.769)[a] | 0.001* |
| Uremic signs and symptoms | | | |
| Yes | 1 | 1 | |
| No | 0.532 (0.079–3.57) | 0.092 (0.009–0.955)[a] | 0.04* |
| Using of anticoagulant and/or antiplatelet drug | | | |
| Yes | 1 | 1 | |
| No | 1.156 (0.168–7.936) | 0.017 (0.001–0.446)[a] | 0.014* |

Reference Variable: No clotting/bleeding.

*Variables that showed a significant association, P < 0.05.

[a]Variable category with P <0.05.

A similar result was reported in the United States of America, that evaluated clotting of the HD circuit in general care patients and critically ill patients. The overall rate of extracorporeal circuit clotting was 5.2% in all sessions [12]. Circuit clotting occurred in 2.4% of HD sessions with heparin, according to a retrospective study comparing two intra-dialytic heparin protocols: "routine heparin-use" during HD (routine heparin prime/bolus dose) and heparin-free HD (saline prime, heparin avoidance). However, heparin-free HD is associated with a higher rate of HD circuit clotting (9.1%) [13]. In contrast, other studies reported a higher rate of circuit clotting during HD. A study conducted in Brazil showed that filter clotting occurred in 19 patients (25.3%) and 29 sessions (14.9%) [19]. This study evaluated and compared intra- and post-dialysis complications in critically ill AKI patients undergoing extended daily dialysis sessions of different durations (6 versus 10 hours), which could explain the higher rate of HD filter clotting events, and the fact that many critically ill patients develop hemostatic abnormalities in addition to the longer dialysis duration.

Additionally, a retrospective cohort study conducted in Belgium found that circuit coagulation was reported in 17.5% of the sessions. This higher rate of clotting events could be attributed to the absence of systemic anticoagulation [20]. Moreover, this study conducted a retrospective analysis of HD sessions in intensive care unit patients, which could be an additional factor contributing to clotting occurrences. Patients admitted to the intensive care unit and requiring dialysis for AKI often present with a systemic inflammatory state [21], which is known to be associated with the activation of coagulation pathways [22].

Lower clotting of the HD circuit incidence was reported in a retrospective study conducted in the USA (1% of 400 HD treatments) [23]. This may be attributed to the operational definition of clotting used in this study. The researchers defined clotting as complete clotting that required the replacement of the blood tubing and dialyzer to complete treatment. The higher blood flow rate (378 ± 46 mL/min) and more aggressive normal saline flushing of the circuit employed in this protocol could also explain the lower rate of clotting. Moreover, there might be underreporting of clots due to the retrospective study design, in which existing data were extracted from chart reviews. Another study from Spain found that 1.9% of patients developed clotting, which is lower than our finding (19.4% of patients) with a higher mean blood pump flow (346 ± 47 ml/min), given a higher risk of coagulation from a lower pump flow [17].

In the current study, 15.4% of patients developed bleeding, and this finding was higher than a study conducted in Spain (4.4%) [17]. This variation might be due to the difference in the study design and setting, even though the study reported that oral anticoagulants were administered to 18.4% of patients, which by itself is thought to increase the risk of bleeding. However, in a Spanish study, each patient was asked to report any bleeding or thrombotic complications that arose in the week before data collection.

In the present study, it was observed that anticoagulation was achieved with UFH or saline flushes of 30 to 50 ml administered every 30 min if UFH was contraindicated. There were three strategies regarding UFH order for HD in current study settings, and each patient was dialyzed using any of these anticoagulation methods. The first is the standard dosing regimen of UFH, consisting of a bolus of 1250–2000 U, followed by a continuous infusion of 1000–1250 U/h. The second was the minimal/half-dose method with a 625–1000 bolus dose and 500–625 infusion rate. Heparin-free dialysis was also the third method used in patients who were actively bleeding or were at an increased risk of bleeding. In our study, heparin-free hemodialysis was performed in 384 sessions due to contraindications to heparin, such as elevated bleeding risk, active hemorrhage, or a history of heparin-induced thrombocytopenia.

Due to its brief half-life and extensive medical usage over time, UFH has emerged as the predominant anticoagulant for HD [9] Nonetheless, the dosage administered shows significant divergence. This study reveals that the average total UFH dose throughout HD sessions was 5000 ± 1000 units, with heparin infusion being discontinued one hour before dialysis completion in the majority (87.2%) of cases to reduce the potential for bleeding from the access site post-needle withdrawal.

In the present study, extracorporeal circuit clotting was assessed by visual inspection and noted by nurses. It is characterized by extremely dark blood or black striations in the tube and a sudden rise in pressure readings. They only recorded clotting as being present or absent without grading it in line with several studies done elsewhere [13,17,19,24]. In contrast, other studies have reported the degree of clotting by classifying it as mild/slight, moderate, and severe.

HD is often performed three times a week for three to four hours each session; however, the length of these sessions varies from patient to patient. One consequence of intradialysis, circuit clotting, includes treatment interruption and patient blood loss, which may exacerbate hemodynamic instability [19]. In 1.7% of all sessions with clotting, dialysis was stopped early without the possibility of retransfusing blood from the extracorporeal circuit; hence, the bloodline was discarded. However, in 0.2% of the sessions, dialysis was terminated early by the return of blood.

According to a study from Belgium, clotting reduced the length of time spent receiving dialysis in 15.2% of sessions, completely blocking the circuit, and preventing retransfusion in 4.2% of sessions, which is greater than our findings [20]. These variations may be due to differences in dialysis protocols. In contrast to our study, the study combined a citrate-enriched dialysate with a heparin-coated dialyzer with no systemic UFH. Similarly, a study conducted in the USA comparing two intradialytic heparin protocols reported that in a heparin-free HD protocol, circuit clotting resulted in a change of the extracorporeal circuit in 7.3% of sessions. However, this was required in only 0.8% of HD sessions with heparin, and early termination of HD (1.6%) was similar to that in this study [13]. Bleeding is another important intradialysis complication that might require adjusting the dose of UFH (1.6% of all sessions with bleeding) or blood transfusion (0.2%). Another study revealed that patient weight, circuit clotting, and bleeding of the vascular access after disconnection were the most frequently employed techniques for changing the dosage [17].

Identifying predictors of anticoagulation-related outcomes during HD in patients with AKI is important. Identifying and preventing patients at risk of such complications can increase practitioner awareness of HD-related complications. Similar to other studies [12,13,20,23], the blood flow rate showed a statistically significant association with the occurrence of circuit clotting. In this study, lower delivered blood flow rates were associated with higher circuit-clotting rates. Our study identified that the primary cause of reduced blood flow in patients experiencing clotting was multifactorial. Specifically, we observed that hemodynamic instability, often due to underlying conditions such as hypotension or fluid overload, significantly reduced dialysis treatment tolerance. This instability can lead to fluctuations in blood flow rates, which may increase the risk of clot formation. Additionally, we found that issues related to vascular access, including stenosis or thrombosis within the access site, further limited blood flow rates. These complications can create a vicious cycle where reduced flow promotes clot formation, exacerbating the initial problem. Additionally, as the total number of HD sessions increased, the probability of circuit clotting increased, unlike in a study conducted in Belgium [20]. This difference could be due to variations in their research methodology and the inclusion of study participants with varying levels of clotting risk. Patients who did not receive prophylactic or therapeutic anticoagulants and/or antiplatelet drugs were less likely to develop bleeding compared to those taking these drugs. Similarly, a study from Spain found that bleeding complications were more frequent in patients receiving oral anticoagulants [17]. Consistent with previous studies, age and sex were not associated with bleeding events [13,17].

However, there were some limitations to our study. Bleeding was reported only at the time of the dialysis session. Therefore, this may not provide a comprehensive picture of the bleeding risk associated with dialysis. Additionally, the study did not exclude patients with comorbidities (such as deep vein thrombosis, active malignancy, severe heart failure, or severe liver disease). These conditions can impact anticoagulation outcomes and may confound the interpretation of results.

## Conclusions

In conclusion, clotting was observed in 19.4% of the participants in the study, while bleeding was reported in 15.4% of them. These findings highlight the importance of thorough monitoring due to the complications resulting in treatment interruptions and blood loss. The study identified a significant association between circuit clotting and variables such as blood flow rate and the frequency of HD sessions. Furthermore, it revealed a notable occurrence of bleeding during HD, mainly manifesting as minor cases. Individuals who received prophylactic or therapeutic anticoagulants and/or antiplatelet medications were more susceptible to bleeding compared to those who did not undergo these treatments.

## Acknowledgments

The authors thank the School of Pharmacy, College of Health Sciences, Addis Ababa University, and the nursing staff at TASH and SPHMMC dialysis clinics, who were very cooperative and helpful in providing constant assistance.

## Author Contributions

**Conceptualization:** Hanan Muzeyin Kedir, Abdella Birhan Yabeyu, Addisu Melkie Ejigu, Tamrat Assefa Tadesse, Eskinder Ayalew Sisay.

**Data curation:** Tamrat Assefa Tadesse.

**Formal analysis:** Hanan Muzeyin Kedir, Abdella Birhan Yabeyu, Eskinder Ayalew Sisay.

**Investigation:** Hanan Muzeyin Kedir, Addisu Melkie Ejigu, Eskinder Ayalew Sisay.

**Methodology:** Hanan Muzeyin Kedir, Abdella Birhan Yabeyu, Tamrat Assefa Tadesse.

**Software:** Hanan Muzeyin Kedir, Abdella Birhan Yabeyu.

**Supervision:** Hanan Muzeyin Kedir, Addisu Melkie Ejigu, Tamrat Assefa Tadesse, Eskinder Ayalew Sisay.

**Visualization:** Eskinder Ayalew Sisay.

**Writing – original draft:** Hanan Muzeyin Kedir, Abdella Birhan Yabeyu, Addisu Melkie Ejigu, Tamrat Assefa Tadesse, Eskinder Ayalew Sisay.

**Writing – review & editing:** Hanan Muzeyin Kedir, Abdella Birhan Yabeyu, Tamrat Assefa Tadesse, Eskinder Ayalew Sisay.

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
