## [Decision Letter · Decision Letter 0]

23 Sep 2024

PONE-D-24-07202Anticoagulation-related complications and their outcomes in hemodialysis patients with acute kidney injury at selected hospitals in Ethiopia.PLOS ONE

Dear Dr. Kedir,

Thank you for submitting your manuscript to PLOS ONE. After careful consideration, we feel that it has merit but does not fully meet PLOS ONE’s publication criteria as it currently stands. Therefore, we invite you to submit a revised version of the manuscript that addresses the points raised during the review process.

We look forward to receiving your revised manuscript.

Kind regards,

Peter R. Corridon

Academic Editor

PLOS ONE

2. We note that your Data Availability Statement is currently as follows: [All relevant data are within the manuscript and its Supporting Information files.] Please confirm at this time whether or not your submission contains all raw data required to replicate the results of your study. Authors must share the “minimal data set” for their submission. PLOS defines the minimal data set to consist of the data required to replicate all study findings reported in the article, as well as related metadata and methods (https://journals.plos.org/plosone/s/data-availability#loc-minimal-data-set-definition). For example, authors should submit the following data: - The values behind the means, standard deviations and other measures reported; - The values used to build graphs; - The points extracted from images for analysis. Authors do not need to submit their entire data set if only a portion of the data was used in the reported study. If your submission does not contain these data, please either upload them as Supporting Information files or deposit them to a stable, public repository and provide us with the relevant URLs, DOIs, or accession numbers. For a list of recommended repositories, please see https://journals.plos.org/plosone/s/recommended-repositories. If there are ethical or legal restrictions on sharing a de-identified data set, please explain them in detail (e.g., data contain potentially sensitive information, data are owned by a third-party organization, etc.) and who has imposed them (e.g., an ethics committee). Please also provide contact information for a data access committee, ethics committee, or other institutional body to which data requests may be sent. If data are owned by a third party, please indicate how others may request data access.

3. Please upload a copy of Figure 1, to which you refer in your text on page 15. If the figure is no longer to be included as part of the submission please remove all reference to it within the text.

Additional Editor Comments (if provided):

Reviewers' comments:

Reviewer's Responses to Questions

**Comments to the Author**

1. Is the manuscript technically sound, and do the data support the conclusions?

Reviewer #1: Partly

Reviewer #2: Yes

2. Has the statistical analysis been performed appropriately and rigorously? 

Reviewer #1: Yes

Reviewer #2: Yes

3. Have the authors made all data underlying the findings in their manuscript fully available?

Reviewer #1: Yes

Reviewer #2: Yes

4. Is the manuscript presented in an intelligible fashion and written in standard English?

Reviewer #1: Yes

Reviewer #2: Yes

5. Review Comments to the Author

Reviewer #1: I read with interest the paper by Kedir et al. about bleeding and clotting complications in HD patients treated for AKI. The paper has some limitations, mainly due to the challenging and different settings than usually observed in Western countries; I only suggest some revisions to improve the comprehension and generalizability of study results.

- The anticoagulation strategies greatly vary between centers; please indicate how many patients do not receive heparin, the reason, and the outcomes in this subgroup. At the same time, the authors can state the reason for the unavailability of citrate anticoagulation (difficulty in supplying? Costs? Both?) considering the advantages in AKI patients (see as example 10.3390/biomedicines11092570)

- The correlation between blood flow rate and clots may be direct (e.g., hemodynamic problems in the patients with low tolerance to the treatment) or indirect (e.g., problems in the vascular access limiting the flow rate favoring clot formation). Please specify the primary cause of reduced blood flow in patients with clotting and comment on it.

- If I understand correctly, all patients have central venous catheters as vascular access. Please include the characteristics of the CVC (materials, tunneled or not, temporary or permanent) and the site (jugular/femoral vein) because all these characteristics may influence the clotting risk. If possible, mention the machinery and technique adopted for the dialysis sessions (HD, HDF, hemofiltration).

Reviewer #2: The manuscript by Kedir et al. “Anticoagulation-related complications and their outcomes in hemodialysis patients with acute kidney injury at selected hospitals in Ethiopia” looks interesting. This article aims to evaluate anticoagulation outcomes and related factors in hemodialysis (HD) patients with acute kidney injury (AKI) at selected hospitals in Ethiopia. Descriptive statistics were utilized to summarize the data from AKI patients undergoing HD during October 2021 to March 2022, and multinomial logistic regression analysis was employed to identify factors associated to clotting and bleeding. The authors found that clotting affected 19.4% of participants, while bleeding occurred in 15.4%, underscoring the importance of close monitoring.

The manuscript is well written and provides enough strategies to fulfill its aims, though language may need some polishing for better flow. In conclusion, I support publication of this work and believe that it would be interesting for researchers working in the field.

6. PLOS authors have the option to publish the peer review history of their article (what does this mean?). If published, this will include your full peer review and any attached files.

Reviewer #1: No

Reviewer #2: No

---

## [Author Response · Author response to Decision Letter 0]

4 Nov 2024

Dear academic editor, Thank you for your guidance. We have reviewed and updated the manuscript to ensure full compliance with PLOS ONE's style requirements, including those for file naming.

Dear Academic Editor, thank you for the clarification on the Data Availability Statement. We confirm that our submission includes all data necessary to replicate the findings of our study,

Dear Academic Editor, thank you for bringing this to our attention. We have uploaded Figure 1 in the revised manuscript as referenced in the text on page 15.

Dear Academic Editor, thank you for the reminder regarding the reference list. We have carefully reviewed all references to ensure they are complete, accurate, and up-to-date.

Dear reviewer, thank you for your feedback. Our study was conducted with rigorous scientific standards, including a multicenter observational design, consistent protocols across 1010 HD for 175 AKI patients, and real time data collection by trained personnel. We used multinomial logistic regression and descriptive statistics to identify key associations, such as between HD frequency, blood flow rates, and anticoagulation complications. Our conclusions are directly drawn from these findings, aligning statistical results with observed outcomes. We believe this demonstrates the technical soundness and data support for our conclusions.

Thank you for your positive feedback regarding the statistical analysis.

We appreciate your acknowledgment of the data availability.

Thank you for your positive feedback regarding the clarity and language of our manuscript.

Dear Reviewer 1, Thank you for your valuable comments. We have updated the Characteristics of HD Sessions section to include the number of patients who underwent heparin-free dialysis, along with the reasons for it, such as elevated bleeding risk or contraindications. 

Citrate anticoagulation is not a common method of anticoagulation in the Ethiopian setting due to its limited availability, the need for regular and strict monitoring parameters, and our resource-constrained environment, despite its advantages for patients with AKI.

Thank you for your insightful feedback regarding the correlation between blood flow rate and clot formation in patients undergoing dialysis for acute kidney injury. We have clarified in our revised manuscript to emphasize both the direct and indirect factors contributing to reduced blood flow in this patient population. Thank you again for your valuable input.

Thank you for your comments. We have included details about the central venous catheters (CVCs) used in our study in the revised manuscript. Specifically, we utilized non-tunneled (temporary) hemodialysis catheters placed in the jugular or femoral vein sites.

Thank you for your positive feedback regarding the technical soundness of our manuscript.

Thank you for your positive feedback regarding the statistical analysis.

We appreciate your acknowledgment of the data availability.

Thank you for your positive feedback regarding the clarity and language of our manuscript.

Dear Reviewer 2, Thank you so much for your thoughtful review of our manuscript. We truly appreciate your positive feedback on the study’s relevance and the statistical analyses we employed. We acknowledge your comment about the language needing some polishing for better flow. We have carefully revised the manuscript to enhance clarity and readability throughout the text.

---

## [Decision Letter · Decision Letter 1]

18 Nov 2024

Anticoagulation-related complications and their outcomes in hemodialysis patients with acute kidney injury at selected hospitals in Ethiopia.

PONE-D-24-07202R1

Dear Dr. Kedir,

We’re pleased to inform you that your manuscript has been judged scientifically suitable for publication and will be formally accepted for publication once it meets all outstanding technical requirements.

Kind regards,

Peter R. Corridon

Academic Editor

PLOS ONE

Additional Editor Comments (optional):

Reviewers' comments:

Reviewer's Responses to Questions

**Comments to the Author**

1. If the authors have adequately addressed your comments raised in a previous round of review and you feel that this manuscript is now acceptable for publication, you may indicate that here to bypass the “Comments to the Author” section, enter your conflict of interest statement in the “Confidential to Editor” section, and submit your "Accept" recommendation.

Reviewer #1: All comments have been addressed

2. Is the manuscript technically sound, and do the data support the conclusions?

Reviewer #1: Yes

3. Has the statistical analysis been performed appropriately and rigorously? 

Reviewer #1: Yes

4. Have the authors made all data underlying the findings in their manuscript fully available?

Reviewer #1: Yes

5. Is the manuscript presented in an intelligible fashion and written in standard English?

Reviewer #1: Yes

6. Review Comments to the Author

Reviewer #1: The Authors have revised the manuscript according to all suggested comments. All primary issues have been addressed, especially considering the specific geographical area and related problems. In my opinion, no further comments are requested.

7. PLOS authors have the option to publish the peer review history of their article (what does this mean?). If published, this will include your full peer review and any attached files.

Reviewer #1: No

---

## [Editor Report · Acceptance letter]

21 Nov 2024

PONE-D-24-07202R1 

PLOS ONE

Dear Dr. Kedir, 

I'm pleased to inform you that your manuscript has been deemed suitable for publication in PLOS ONE. Congratulations! Your manuscript is now being handed over to our production team.

Kind regards, 

on behalf of

Dr. Peter R. Corridon 

Academic Editor

PLOS ONE